# Reproducibility report: Interpretable Complex-Valued Neural Networks For Privacy Protection

**University of Amsterdam (UvA)**
**Master's Artificial Intelligence**

**Luuk Kaandorp** (11992190), **Ward Pennink** (11306246)
**Ramon Dijkstra** (12017663) and **Reinier Bekkenutte** (13438557)

## Reproducibility Summary

**Scope of Reproducibility**

In this reproducibility report, the following two main claims of Xiang et al. (2020)'s paper are tested:

- The performance of a Deep Neural Network (DNN) is largely preserved when comparing DNNs with complex encoded features to DNNs with non-encoded features.

- It is more difficult for an attacker to reconstruct the original input from the complex encoded features as compared to the non-encoded features.

**Methodology**

Since the code was not made publicly available we implemented our own version of the reported DNNs. Baseline DNNs were created using the default model architecture. The figures and math of the original paper were used to recreate the structure of the complex-valued DNNs, in which the model is divided into an encoder, a processing module on the cloud, and a decoder. The goal of the complex-valued DNN is to make sure that the features are rotated and obfuscated to ensure that the privacy of the data is secured. We compare the performance of the baseline and complex-valued DNNs. Then, we test the robustness of the models against privacy attacks, where potential attackers were mimicked using inversion attacks.

**Results and Discussion**

Overall, our results are not in line with the results of the original paper. We were not able to reach the performance of the baseline models that is obtained in the original paper. Additionally, our complex models obtain a much higher classification error than the baseline models, as opposed to the claim that the performance is largely preserved. Regarding the second claim, however, we did find evidence supporting the claim. The attacker trying to reconstruct the intermediate-level features had a harder time reconstructing the obfuscated features than it did when the features were left untouched.

**What was easy**

Creating the baseline DNNs and the inversion attacker was relatively easy because much of the code and additional information about these models can be found online. Additionally, how to create the complex DNNs by dividing the baseline DNNs into an encoder, processing module and decoder was clearly described in the original paper. Furthermore, the mathematics behind the overall optimization function of the proposed complex model was described in a clear way.

**What was difficult**

The main difficulty arose from the fact that certain hyperparameters and model structures were not specified by the authors. This mainly led to problems with the implementation of the GAN-based encoder and the rotating of the features. Additionally, the complex tensor support in PyTorch caused a few problems in the backpropagation stage of the training process. Lastly, some of the modified layers for the complex-valued DNNs had no clear mathematical formulations. The combination of the previously mentioned factors made it difficult to reproduce the original results.

# 1  Introduction

Nowadays, Deep Neural Networks (DNNs) are used in many applications for processing large quantities of data and for inferring relevant information from this data. Most of this data is collected from smaller, less powerful devices like mobile phones and Internet of Things (IoT) devices. These devices often do not have the processing power to apply a DNN in real-time. Therefore, much of the processing load is offloaded to the cloud.

However, offloading data to the cloud introduces privacy and confidentiality risks, as private data is sent over the internet. Attackers who can intercept the data might be able to reconstruct and infer privacy sensitive information from individuals.

This calls for better encryption in DNN applications where processing is offloaded to the cloud. The paper by Xiang et al. (2020) proposes a new model architecture that encrypts the data in complex-valued and rotated features, to prevent an attacker from inferring or reconstructing privacy sensitive information.

# 2  Scope of reproducibility

The paper by Xiang et al. (2020) looks at the privacy problems that are associated with running data processing operations on the cloud. To combat the privacy risks associated with these cloud-based operations, the authors propose a method to encrypt the real features in complex encoded features as they are being passed through the model on the cloud.

In this reproducibility report, the following main claims of the paper are tested:

- The performance of a Deep Neural Network (DNN) is largely preserved when comparing DNNs with complex encoded features to DNNs with non-encoded features.

- It is more difficult for an attacker to reconstruct the original input from the complex encoded features as compared to the non-encoded features.

# 3  Methodology

There was no original code of the authors available. Therefore, we coded our own models in Python using the PyTorch library. Our implementation is described in detail in the following subsections.

## 3.1  Model descriptions

### 3.1.1  Baseline DNNs

The original paper uses a range of well-known DNNs to test their claims. These models have several improved variations where additional layers are added to increase their respective performance. However, the authors do not specify which architecture of the models with corresponding hyperparameters they use to obtain their results. Therefore, we decided to use the default models for our experiments as specified in their original papers. We used the LeNet model by Lecun et al. (1998), ResNet-56 and ResNet-110 by He et al. (2015), and VGG-16 by Simonyan and Zisserman (2014).

### 3.1.2  Complex DNNs

For the complex DNNs, the baseline DNNs are divided into three parts. The first part is called the encoder and consists of the first few layers of each model. The main task of the encoder is to encrypt the features by applying a transformation that maps each feature to the complex space and rotates it. The rotation angle is saved on the local device to be able to re-rotate the features in the decoder. These encoded features are then sent to the processing module on the cloud, which performs the bulk of the computations. The processing module follows the same architecture as the standard models, albeit with some adjustments in the layers to make them rotation-invariant. Finally, the decoder receives the output from the processing module, re-rotates it and maps it back to the real space. The re-rotated features will go through the last part of the network, which is responsible for the classification of the data.

These three modules are jointly trained in order to preserve the performance on the original task, as well as a way to encode the features in such a way that the privacy is maintained when sending data to the cloud. The overall structure of the three modules for each DNN is visualized in Figure 1. This structure is based on the defined implementation details from the original paper. For the ResNet models, the $\alpha$ variant is implemented. From now on, we just call these ResNet.

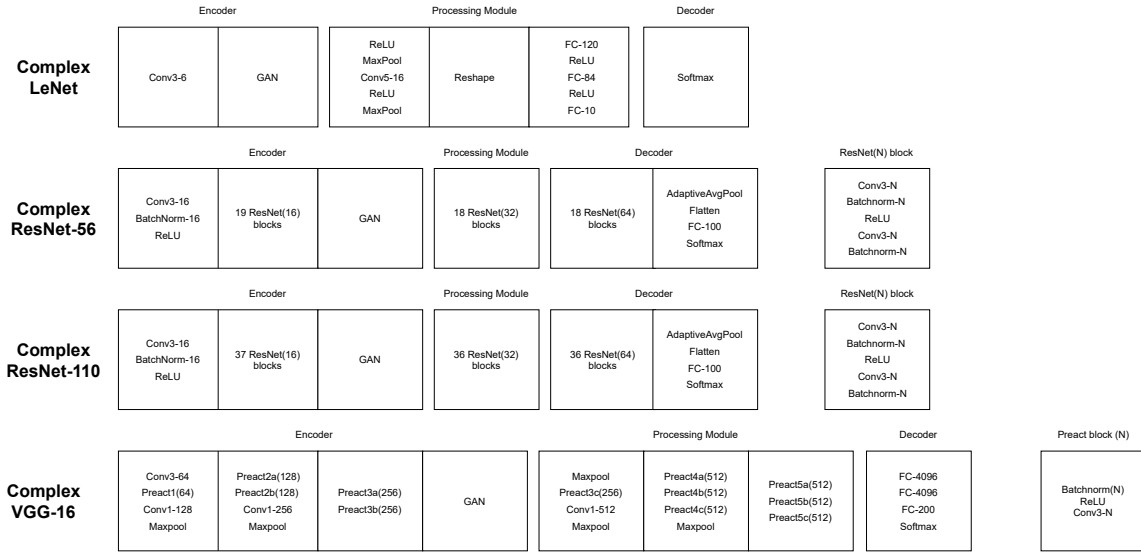

Figure 1: Complex DNN architecture.

### 3.1.3 GAN based encoder

The implementation of the GAN based encoder is adapted from the original paper. The generator of the GAN first rotates the original input feature $a$ and maps it to the complex space. This rotation is done by rotating each feature with a distinct angle $\theta$. This angle is stored in the model to be able to reverse the rotation in the decoder. Additionally, a random tensor $b$ with a comparable magnitude to $a$ is generated. This tensor is added to the rotation to serve as a fooling counterpart. Together, these transformations constitute the encoding of the input feature. Then, for each feature, $k-1$ different $\theta$s are generated from a uniform distribution $(0, \pi)$ to create $k-1$ fake features. Both the real and fake features are then fed into the discriminator to optimize the generator through adversarial learning. The GAN based encoder thus introduces the encryption that ensures the data's privacy.

### 3.1.4 Attackers

The original paper tests the robustness of the model against two different attackers. To validate the privacy claim made by the paper, we implement one of these attackers, i.e. the inversion attacker. The inversion attack attempts to reconstruct the original input image from the intermediate features using a U-net model (Ronneberger et al., 2015). This attack is executed on the processing module. Due to time constraints, we decided to implement inversion attacker 2 from the original paper. We followed the structure of the original U-net with 8 blocks. The input was first up-sampled to the size of the images using an up-sample layer. The pixel values of the input were then scaled to [0,1] from the original [0,255] to make use of the Mean Squared Error (MSE) loss. The model consists of four down-sampling blocks and four up-sampling blocks, which respectively reduce or expand the size of the feature by a factor of 2. The eventual output of this model is of the same size as the input and the original images. When testing the performance on the encoded features, the complex features are first cast to the real space.

### 3.2 Datasets

The DNNs were trained, evaluated, and tested on one of the following three datasets.

- **CIFAR-10**: This dataset consists of 60,000, 32x32 RGB images, each annotated with one of 10 balanced classes. 50,000 images were used as training data, 5,000 were used for validation, and the remaining 5,000 were used for testing. The CIFAR-10 dataset was used to train and test the LeNet model described above. In the preprocessing step, all the images are normalized. The dataset is publicly available at https://www.cs.toronto.edu/ kriz/cifar.html.

- **CIFAR-100**: This dataset, similar to CIFAR-10 consists of 60,000, 32x32 RGB images. This time it is annotated with one of 100 balanced classes. Each class has 500 training images, 50 validation images, and 50 images to test on. In the preprocessing step, all the images are normalized. The dataset is publicly available at https://www.cs.toronto.edu/ kriz/cifar.html.

- **CUB-200**: This dataset is a collection of 11,788 RGB pictures of different kinds of birds. Each bird belongs to one out of 200 species, which is the target that is learned in this dataset. The train, validation, and test split were created with the following ratios respectively: 0.5, 0.25, and 0.25 (Wah et al., 2011). Because the dataset contains images of different sizes, in the preprocessing step each image was resized to a width and height of 56 pixels before it was normalized. The dataset is publicly available at https://deepai.org/dataset/cub-200-2011.

### 3.3 Hyperparameters

When specified, we follow the hyperparameters provided by the original paper. However, no values were given for the $k$ that is used in the training of the encoder, the optimizer, or the learning rate. Therefore, their optimal values were examined via a grid search on a predefined range of values for the complex LeNet. The model is trained until convergence on the CIFAR-10 dataset. The results are summarized in table 1. Because there are only very minor differences between the different hyperparameter configurations, we settled on using a value of $2$ for $k$. Additionally, we chose the Adam optimizer as it is generally considered to be a good optimizer for DNNs and we set the learning rate at $3e^{-4}$.

| Value of k | Learning rate | Training loss | Train accuracy | Test accuracy |
|:----------:|:-------------:|:-------------:|:--------------:|:-------------:|
| 2 | 3e-4 | 1.50 | 0.54 | 0.50 |
| 2 | 1e-3 | 1.78 | 0.52 | 0.49 |
| 4 | 3e-4 | 1.72 | 0.53 | 0.49 |
| 4 | 1e-3 | 1.69 | 0.54 | 0.49 |
| 10 | 3e-4 | 1.94 | 0.58 | 0.48 |

Table 1: Results after convergence for training the complex LeNet version on CIFAR-10 with different learning rates and values of $k$.

### 3.4 Implementation details

In this section, we briefly describe any choices or assumptions we made during the implementation process. Since not all hyperparameters and model architectures were clearly specified in the original paper, this section contains important details that give others the opportunity to reproduce our results.

#### 3.4.1 GAN

For the GAN, we use the BCEWithLogitsLoss function to calculate the loss and optimize the encoding of the features. The transformation to the complex space is done by adding a complex tensor $b$ to the features that should be of equal magnitude to $a$. We implemented this by sampling $b$ from a unit Gaussian distribution and scaling it to have a magnitude that is equal to that of $a$. The rotation angle $\theta$ was then sampled from a uniform distribution between $0$ and $2\pi$. The fake angles $\Delta\theta$, which are meant to trick the discriminator, are sampled from a uniform distribution between $0$ and $\pi$. Additionally, in the discriminator of the GAN, a linear layer has been added to be able to map the output to a binary value, indicating whether a feature transformation is real or fake.

#### 3.4.2 Models

Our implementation of the baseline models for LeNet and ResNet is equal to the implementation in their original papers. For the VGG16, the original paper mentions multiple variants of the model. We decided to implement the D-variant. For its complex counterpart, it was not clear how the division into encoder, processing module, and decoder was done. The paper mentions that the encoder constitutes all the layers until the last 56x56 feature map. We interpreted this as all convolutional layers until the third Max Pooling layer.

#### 3.4.3 Complex layers

Because we implemented the complex models using PyTorch, we had to separate the features into a real and imaginary part to ensure that the features can be correctly passed through the processing module. These parts are passed through each of the complex layers and activation functions independently, before they are combined into one complex tensor again.

The original paper specifies the complex convolutional layer as a default convolution where the bias term is set to false. However, due to the limited Pytorch support of complex tensors, we had to write our own convolution function. This complex convolution function calculates the real output by subtracting the imaginary feature from the real feature after

they are both passed separately through a convolutional layer without a bias term. The imaginary output is calculated by not subtracting but by summing these two terms. The complex ReLU, complex batchnorm, and complex MaxPool are implemented using the formulas in the original paper. To calculate the norm that occurs in these formulas, we first recombine the real and complex part of the feature, after which we take the square root of the squared, combined tensor.

### 3.4.4 Attackers

During the implementation of the inversion attacker, a few assumptions had to be made. First of all, it was not specified on which data the attacker model was trained. We assumed that the attacker model was trained on feature data, i.e. that it had learned to reconstruct the original input data when given a feature map. Therefore, the U-Net was trained on data that had already passed through the encoder layers of the models, but was not encrypted using the GAN. In our results, we compare the performance of this pretrained U-Net on reconstructing the un-encoded features versus the encoded features.

## 3.5 Experimental setup and code

The models in this report are implemented in Python using the PyTorch Lightning framework. All datasets are split into three separate sets as described in section 3.2. The validation set is used to dynamically determine the number of epochs that the model is trained on by checking whether the validation loss has converged. The model has converged after the validation loss does not decrease for more than 10 epochs. The parameters of the model at the epoch with the lowest validation loss are then saved as final model.

Both the baseline and the complex models are evaluated on two metrics: the performance of the model in terms of the classification error and the robustness of the model against the inversion attack. The robustness against the inversion attack is measured by the reconstruction error, which is calculated using the Mean Squared Error Loss on the reconstruction of normalized images.

All code for the implementation of the different models has been made publicly available on https://github.com/Ramonprogramming/AI-FACT. The code has been written with a strong focus on modularity, which makes it easy to vary between the different models. Additionally, the pre-trained models that have been used for obtaining the results have been uploaded to the Github repository.

## 3.6 Computational requirements

The computational requirements of the experiments are described in the following subsections.

### 3.6.1 Hardware description

We ran our experiments on a Windows 10 desktop pc with an Intel Core i5-8500K @ 3.6GHz CPU, Nvidia GTX 1080 Ti 11GB GPU, and 16GB RAM.

### 3.6.2 Model run-time

We have measured the total time it takes for the models to train until convergence. Our convergence criterium measures the validation accuracy and includes a minimum delta of 0 and a patience of 10. Therefore, the models are still trained for 10 epochs after it has converged. The results are displayed in table 2.

| Model | Dataset | Number of epochs | Batch size | Run-time |
|---|---|---|---|---|
| LeNet | CIFAR-10 | 96 | 512 | 00:18:01 |
| Complex LeNet | CIFAR-10 | 61 | 512 | 00:16:56 |
| ResNet-56 | CIFAR-100 | 13 | 512 | 00:10:06 |
| Complex ResNet-56 | CIFAR-100 | 17 | 512 | 00:12:57 |
| ResNet-110 | CIFAR-100 | 8 | 512 | 00:12:17 |
| Complex ResNet-110 | CIFAR-100 | 22 | 512 | 00:33:29 |
| VGG-16 | CUB-200 | 40 | 256 | 00:38:27 |
| Complex VGG-16 | CUB-200 | 1 | 256 | 00:10:41 |

Table 2: Run-times of training and testing the different models.

# 4 Results

The results section covers the results of the experiments we conducted to validate the claims made in the paper. The two subsections each address one of the claims.

## 4.1 Results performance claim

This section covers the results of our experiments regarding the first claim that was made in the paper. This claim states that the accuracy is largely preserved when encoding the model's features. In our experiment, we compare the complex models with their baseline counterparts based on the classification error rate on a separate test set. All models have been trained till convergence. Additionally, the results from the original paper are shown to be able to directly compare the results. The results are summarized in table 3.

| Model | Original paper results | | Reproduced results | |
|---|---|---|---|---|
| | **Baseline DNN** | **Complex DNN** | **Baseline DNN** | **Complex DNN** |
| LeNet | 19.78 | 17.95 | 36.92 | 49.56 |
| ResNet-56 | 53.26 | 44.37 | 70.48 | 85.00 |
| ResNet-110 | 50.64 | 50.94 | 73.00 | 88.88 |
| VGG-16 | 56.78 | 78.50 | 23.86 | 99.38 |

Table 3: Classification error rates for standard and complex models.

## 4.2 Results privacy claim

This section looks at the results of our experiments regarding the second claim that was made in the paper. This claim states that adversaries have more difficulty inferring inputs from the complex models than the standard models. In our experiment, we compare the complex models with their default counterparts in terms of the classification error rate of the attacker. The results are summarized in table 4.

| Model | Original paper results | | Reproduced results | |
|---|---|---|---|---|
| | **Original DNN** | **Complex DNN** | **Original DNN** | **Complex DNN** |
| LeNet | 0.0769 | 0.2353 | 0.2317 | 0.4484 |
| ResNet-56 | 0.0929 | 0.2473 | 0.2054 | 0.3698 |
| ResNet-110 | 0.1050 | 0.2419 | 0.2257 | 0.3230 |

Table 4: Adversary reconstruction error rates for standard and complex models.

# 5 Discussion

As visible in the previous results section, we were not able to reproduce all claims made by the authors of the original paper. The main difference we observed in our results regards the performance claim. Whereas in the original paper the performance of the complex models was approximately equal and for some models even better than their baseline counterpart, our complex models reached a significantly lower performance than the baseline models. This difference can be attributed to various reasons. First of all, we were not able to perform an extensive hyper-parameter search and fully optimize the performance of our baseline models. This was partly due to time constraints, but also partly because the original paper did not specify with sufficient detail how their baseline models were implemented. Therefore, our baseline models performed worse, and it could be the case that, by extension, our complex models performed even worse. However, this does raise additional issues. After all, this indicates that the transformation of a normal model to a complex-valued model would only work for highly optimized models, which is not ideal.

Another reason for the large differences in performance could be that the implementation of the complex layers in the paper varied slightly from our implementation. Once again, a clearer description in the original paper concerning the implementation of the models could have prevented this issue. During the implementation of the complex models, we were forced to make various assumptions about how the features could be correctly passed through the model layers. Additional information about the frameworks that were used to obtain the original results would have severely aided our reproducibility study.

The results concerning the privacy claim are much more in line with the original paper. Although we once again do not reach the same reproduction error for the attacker, the overall trend is similar. A clear difference in performance is visible when comparing the baseline model to the complex-valued model. The difference in overall performance can be

explained by minor differences in the implementation of the attacker U-Net model. Therefore, our results support the claim that for complex-valued DNNs it is more difficult for attackers to reconstruct original input data from intermediate level features.

With the provided details in the paper, we did not succeed in reproducing the results from the original paper. To improve reproducibility, the original authors could add figures and pseudo-code.

## 5.1 What was easy

As mentioned in the methodology section, there is no official or third-party code available for this paper. This meant we had to replicate the code based on the paper. While this was difficult for the most part, the following aspects were relatively simple:

- The U-net implementation of the inversion attacker was sufficiently clearly described so that it was easy to replicate the necessary code for this model.

- The computations needed to effectively rotate the features and map them to the complex space were sufficiently well described in the paper, resulting in relatively few problems implementing it.

- Finding and using the datasets was also doable, especially for the datasets that have their respective data loaders built within PyTorch.

## 5.2 What was difficult

Overall, we found that replicating this paper was a challenge. We found the following aspects especially difficult:

- Complex values are especially difficult to work with since there is no support within PyTorch for complex Tensors. This meant that we manually had to create functions that could deal with complex values. The original paper did not provide a very detailed description of how this is done. This could be because the original authors used a different framework (e.g. TensorFlow).

- We found that the details and descriptions that the original paper provides on the model architectures were too limited to replicate the exact results of the original model. We had to make a lot of choices on which models to use and how to design the models such that they can work with complex values. Therefore, our results are very different from the original paper.

- It took us quite a while to get a good grasp of the overall architecture of the complex models proposed by the authors. This was partly due to the two losses that were used, as well as the fact that next to learning the task, the model incorporated learning a GAN to choose the initial features to be encoded.

## 5.3 Communication with original authors

There has been no communication with the original authors. We attempted to reach the authors via e-mail and asked for general tips on reproducing their paper and their implementation of the encoder and complex features. However, we have not received a response.

# 6 Conclusion

This report investigates the reproducibility of the paper 'Interpretable Complex-Valued Neural Networks For Privacy Protection' by Xiang et al. (2020). The paper proposes a new method of creating complex-valued DNNs, for which the privacy of the model is better protected against potential attackers while the performance of the models is largely preserved. We tested both of these claims by creating baseline DNNs and complex-valued DNNs whose performance was evaluated on classifying image data. Additionally, their robustness against inversion attack was measured. The results showed that we were not able to find supporting evidence of the claim that the performance of the complex-valued models is approximately equal to the performance of the baseline models. All our complex-valued DNNs performed worse in comparison to their baseline counterparts. The privacy claim of the model is supported by the results we found. The complex-valued DNNs show an almost twice as high reconstruction error as the baseline models. For the sake of reproducibility, we can conclude that it was hard to reproduce the exact results since certain important implementation details were left out.

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
