# OpenReview forum: "Reproducibility report: Interpretable Complex-Valued Neural Networks For Privacy Protection"
_ML_Reproducibility_Challenge/2020 — Reject_

### Official Review · AnonReviewer1 · 2021-02-09
**Good reproducibility report**

**Rating:** 7
**Confidence:** 4

**Review:**

Even though the authors failed to reproduce results regarding performance, they find that the privacy claim is valid.

They provide a detailed discussion about what was easy and what was difficult when they prepare the reproducibility report. The report is well-written and easy to follow.

**Familiar With The Original Paper:**

I have not read the original paper

**Reproducibility Summary:**

Report has summary

---

### Official Review · AnonReviewer2 · 2021-03-08
**A very good work**

**Rating:** 8
**Confidence:** 3

**Review:**

The authors did well. Though the codes for the original work was not available, they reproduce the work by coding from scratch.

**Familiar With The Original Paper:**

I have not read the original paper

**Reproducibility Summary:**

Report has summary

---

### Official Review · AnonReviewer3 · 2021-03-11
**Accept**

**Rating:** 7
**Confidence:** 2

**Review:**

This paper attempts to replicate the work in Interpretable Complex-Valued Neural Networks For Privacy Protection which in general was not possible due to missing details in the original paper.

The author provided a good description of the work that they have done to replicate the original. The authors identified the details that were missing in the paper. They also mention time constrains, which did not allow them to pursue for example some further hyperparameter tuning.

The authors tried to contact the original authors without any success.

What would have been interesting is to provide a full summary of what would have made the reproduction easier, besides providing the pseudo-code.

All-in-all, it is a good paper.


**Familiar With The Original Paper:**

I have read the original paper

**Reproducibility Summary:**

Report has summary

---

### Decision · Program_Chairs · 2021-03-31

**Decision:**

Reject

**Comment:**

The report is not anonyzimed.
Also, while it is well-written, it doesn't go into as much depth and detail as necessary to understand the results.